# Perceptions and priorities for the development of multiplex rapid diagnostic tests for acute non-malarial fever in rural South and Southeast Asia: An international modified e-Delphi survey

**Rusheng Chew** [1,2,3]*, **Salisa Lohavittayavikant**[1], **Matthew Mayer**[4], **Nicholas Philip John Day**[1,2], **Yoel Lubell**[1,2]

1 Mahidol Oxford Tropical Medicine Research Unit, Bangkok, Thailand, 2 Centre for Tropical Medicine and Global Health, University of Oxford, Oxford, United Kingdom, 3 Faculty of Medicine, University of Queensland, Brisbane, Australia, 4 Demos.Science, Bangkok, Thailand

* chris@tropmedres.ac

## Abstract

### Background

Fever is a common presenting symptom in low- and middle-income countries (LMICs). It was previously assumed that malaria was the cause in such patients, but its incidence has declined rapidly. The urgent need to develop point-of-care tests for the most important causes of non-malarial acute febrile illness is hampered by the lack of robust epidemiological data. We sought to obtain expert consensus on analytes which should be prioritized for inclusion in fingerprick blood-based multiplex lateral flow rapid diagnostic tests (LF-RDTs) targeted towards four categories of patients with acute non-malarial fever in South and Southeast Asian LMICs, stratified by age (paediatric vs. adult) and care setting (primary vs. secondary care).

### Methodology/Principal findings

We conducted a two-round modified e-Delphi survey. A total of 84 panellists were invited, consisting of seven each from 12 countries, divided into three regional panels (Mainland Southeast Asia, Maritime Southeast Asia, and South Asia). Panellists were asked to rank their top seven analytes for inclusion in LF-RDTs to be used in each patient category, justify their choices, and indicate whether such LF-RDTs should be incorporated into algorithm-based clinical decision support tools. Thirty-six panellists (43%) participated in the first round and 44 (52%) in the second. There was consensus that such LF-RDTs should be incorporated into clinical decision support tools. At a minimum, these LF-RDTs should be able to diagnose dengue and enteric fever in all patient categories. There was a clear preference to develop LF-RDTs for pathogens not readily detected by existing technologies, and for direct diagnosis through antigen detection. Pathogen biomarkers were prioritized over host inflammatory biomarkers, with CRP being the only one ranked consistently highly.

**Data Availability Statement:** All relevant data are within the manuscript and its Supporting Information files.

**Funding:** This research was funded in whole, or in part, by the Wellcome Trust [215604/Z/19/Z]. RC was also funded by the UK Government through a Commonwealth Scholarship, and the Royal Australasian College of Physicians through the Bushell Travelling Fellowship in Medicine or the Allied Sciences. The funders had no role in study design, data collection, data analysis, data interpretation or writing of the manuscript.

**Competing interests:** The authors have declared that no competing interests exist.

## Conclusions/Significance

Our results provide guidance on prioritizing analytes for inclusion in context-specific multiplex LF-RDTs and similar platforms for non-malarial acute febrile illness, for which there is an urgent unmet need.

## Author summary

In rural South and Southeast Asia, most acute febrile illness was previously attributable to malaria but the incidence of malaria is declining. To aid diagnosis and prognosis in patients presenting with the common symptom of acute fever with no localising features but in whom malaria has been excluded, there is an urgent need to develop minimally-invasive rapid diagnostic tests (RDTs) which can test for multiple pathogen and host biomarkers. Obtaining expert consensus opinions on what biomarkers these tests should detect will contribute greatly to their development, but there is a paucity of robust epidemiological data on the diverse non-malarial causes of acute fever. We determined the biomarkers which should be included in region-specific fingerprick blood-based RDTs tailored to four patient categories differentiated by age and level of care, in the form of seven-item lists ranked in decreasing order of priority. To provide context for these rank lists, we ascertained the principal factors influencing expert priority-setting and explored perceptions of the clinical utility of such RDTs. Our results provide essential region-specific guidance to aid development of RDTs for acute non-malarial fever, for which there was strong consensus for their inclusion in clinical decision-making tools for low- and semi-skilled healthcare staff.

## Introduction

Infection commonly presents with fever [1], and in regions with high burdens of infectious diseases such as South and Southeast Asia, many patients present acutely with fever but without localising signs and symptoms. It was previously assumed that malaria was the cause in most patients presenting with such fevers, but the roll-out of highly accurate malaria rapid diagnostic tests along with the success of public health efforts to eradicate malaria has resulted in a large decline in malaria incidence [2,3]. Febrile illnesses, however, are still a common cause for seeking medical care in rural communities and, while early intervention is crucial in the management of infection, obtaining clinical and/or microbiological diagnoses is difficult in these resource-poor settings.

Fingerprick blood-based rapid diagnostic tests for several other infectious diseases based on lateral flow technology e.g., HIV and Ebola virus disease, have been developed and rolled out in both high- and low-resource settings [4,5]. These are generally immunoassays based on the detection of microbial antigens and/or antibodies against the causative pathogens, although nucleic acid-based assays have more recently been developed [6]. Similar to these pathogen biomarker assays, tests for several host inflammatory biomarkers of potential diagnostic and/or prognostic significance are also available [7]. The test kits comprise of a recognition layer containing the detection sites, combined with a porous membrane which draws the fluid to be tested past the detection sites by capillary action [8]. They are low-cost, simple to operate, require no specialised equipment, and can be used at the point of care, returning results in 30 minutes or less, thus making them ideal for use in low-resource settings [9,10].

These features, along with the endemicity of myriad infectious diseases in the region, poorly trained health workforce, and lack of diagnostic microbiology capability create a pressing need for multiplex lateral flow rapid diagnostic test kits (LF-RDTs) for multiple infections to guide patient management with the minimum amount of test substrate. The use of multiple single-disease LF-RDTs is unlikely to be cost-effective or logistically feasible, and is sub-optimal for patient comfort. However, ascertainment of the most important infective aetiologies of fever to aid the development of relevant multiplex LF-RDTs and other diagnostic tools in South and Southeast Asia is hampered by the paucity of reliable epidemiological data covering not just incidence, but also disease burden in terms of morbidity and mortality [11]. There is, thus, a high degree of uncertainty in prioritization of pathogens for inclusion in such tests; furthermore, what little published evidence there is comes mainly from cities, rather than rural areas where most of the population live [12]. In addition, there is growing interest in augmenting pathogen-based diagnostics by assaying host biomarkers of inflammation in parallel. These biomarkers can be broadly categorized by use-case. The first of these use-cases is differentiation of viral from non-viral infections, thus improving antimicrobial stewardship [13]. The second is prediction or indication of severe disease, thus improving the identification of patients who require escalation of care [14].

Given the poor understanding of the epidemiology of febrile illness in South and Southeast Asia, we conducted a modified e-Delphi survey to obtain expert consensus on pathogen-specific and host inflammatory biomarker analytes which should be prioritized for inclusion in multiplex LF-RDTs with fingerprick blood as the test substrate, for use in acutely febrile rural residents of South and Southeast Asian low- and middle-income countries (LMICs) who test negative for malaria and in whom the source of infection is unclear. We also aimed to explore the reasoning and contextual background underlying the selection of these analytes.

## Methods

The survey was conducted over two rounds from 20 April to 20 June 2022, and delivered via anonymous web-based questionnaires administered through a bespoke platform. Reminder emails were sent weekly to optimize participation rates. Prior to launch, two experts who would have otherwise been invited as panellists were asked to review the survey to ensure face validity, readability, and usability.

Given the likely spatio-temporal heterogeneity in disease profiles, three regional panels were assembled: South Asia (India, Bangladesh, Pakistan, and Nepal), Mainland Southeast Asia (Thailand, Laos, Myanmar, Peninsular Malaysia, and Cambodia), and Maritime Southeast Asia (Indonesia, Timor-Leste, and the Philippines). Countries were selected to ensure adequate geographical representation, with at least one small country (in terms of relative population size) included per panel.

### Participant selection

A total of 84 experts (seven from each country) were invited to participate in each round. Potential panelists were shortlisted through searches of national infectious diseases and/or tropical medicine specialty society office-bearers; leaders of health-related non-governmental organizations and governmental bodies; researchers from university departments or faculties; and recommendations of experts already selected. The final list was constructed based on work experience, expertise and reputation, involvement in health policymaking, and publication record. To balance perspectives, at least 30% of each regional panel were in non-clinical roles and at least 25% were women. Participants had three weeks to complete each round of the survey, and were able to save and return to their responses if they were unable to complete each round in one sitting.

### First round

In the first round, participants were provided with four scenarios in which they were required to rank seven analytes from a list of pathogen-specific and host inflammatory biomarkers for hypothetical LF-RDTs for acute (≤14 days duration) non-malarial fever of unclear source to be developed for year-round use in each region. The four scenarios corresponded to four patient categories stratified by age and level of care i.e., children (age >28 days and <15 years) and adults (age ≥15 years) seeking healthcare in the community from village health workers or primary health facilities, and being admitted to rural hospitals with limited diagnostic capacity, respectively. Neonates were not included because neonatal febrile illness generally requires assessment and thorough investigation in secondary or higher-level care [15].

Pathogen biomarker options were based on a recent systematic review of published aetiological studies and case reports on non-malarial fever in South and Southeast Asia [16], while host inflammatory biomarker options were selected on the basis of biological plausibility and a non-systematic survey of the literature [17,18]. Participants were able to recommend non-listed analytes as well as refine the provided choices, such as specifying a particular type of antigen, and were asked to explain the reasoning behind their decisions for each scenario. If a host inflammatory biomarker was selected, participants were required to state its perceived utility from the following options: 'differentiation of viral and non-viral fever', 'as a marker of severity', 'both of these reasons', or 'other'.

To assess the perceived value of LF-RDTs in these patient categories, participants were asked whether they would find such a test with a minimum sensitivity of 75% and minimum specificity of 90% across all targets helpful to guide clinical decision-making, and whether they would recommend its inclusion in electronic algorithm-based clinical decision support tools for use in their countries. Finally, to provide further context, participants were asked to rank what they perceived to be the five commonest causes of acute febrile illness in their regions in terms of annual incidence. Consensus was achieved if ≥80% of the participants for each region agreed on a particular response. Questions for which consensus was achieved in the first round were not repeated.

At the conclusion of the first round, participants were invited to provide free-text feedback on the survey structure and questions. Free-text suggestions for improvement were analysed thematically and used to inform the subsequent iteration of the questionnaire. Changes were made based on the number of times an issue was raised, the practicality of the suggested change, and its relevance to the context of the survey.

### Second round

All panellists were re-invited to participate in the second round run two weeks after the end of the first, unless they explicitly declined to participate in the first round. The latter (n = 10) were replaced in the second round, while maintaining the gender and clinician balance described previously.

In this round, only options that had been selected in the first round for each scenario were included. First-round respondents who participated in the second round were shown their previous answers to each question. To assist participants in reaching consensus, the frequencies of every analyte selected in each rank position, along with their sum of weighted scores, were shown graphically. An analyte ranked first was assigned a weighted score seven times more than if it was ranked last. Similarly, for the question on the commonest causes of acute febrile illness, an aetiology ranked first was assigned a weighted score five times more than if it was ranked last. The factors influencing participant reasoning for their rankings from the first round were analysed thematically and presented as discrete options in this round.

In the event that consensus for each rank position was unable to be reached by the end of the second round, agreement was quantified by constructing rank lists of the seven highest-scoring analytes for each scenario and of the five highest-scoring aetiologies of febrile illness were constructed.

Statistical analyses were performed using Microsoft Excel (Microsoft, Washington, USA).

## Results

Thirty-six experts participated in the first round and 44 in the second round, giving overall response rates of 43% and 52%, respectively. Thirty-four of the 36 (94%) first round-respondents also participated in the second round; in this round, regional panel response rates were between 38% and 66%. Participant demographic and professional background details by region for both rounds are shown in Table 1.

### First round

In all panels, there was consensus that such LF-RDTs would help clinical decision-making in all target patient populations, and for the incorporation of these tests into electronic algorithm-based clinical decision support tools for use in their respective countries.

The reasons given by participants for the ordering of their rank lists were distilled into the following statements, which were used as response options in the second round: 'disease prevalence and/or incidence in this age group and care setting', 'potential disease severity and need (or otherwise) for antimicrobial therapy or referral', 'host biomarkers are more useful than aetiological diagnosis in this age group and care setting', 'aetiological diagnosis is more useful than host biomarkers in this age group and care setting', 'pathogens for which there are available RDTs should be prioritised', 'pathogens for which there are no available RDTs should be prioritised', and 'other'.

### Second round

The consensus threshold was not reached for any position in any rank list. Therefore, for each region the seven highest-scoring analytes for each scenario and the five highest-scoring aetiologies of acute febrile illness, based on the sums of their weighted scores, were used to construct the rank lists.

Across all regions, in both primary and secondary care settings and for both paediatric and adult patients dengue NS1 and typhoidal *Salmonella* antigens occupied the top two rank positions (Figs 1–3). This is in keeping with the perception that dengue and enteric fevers are among the top five causes of acute febrile illness in all regions (Fig 4), and with the unsurprising finding that perceived disease incidence and/or prevalence was a primary factor in analyte selection (Fig 5). Also in line with current evidence on the near-elimination of malaria, the disease was thought to be a leading cause of acute febrile illness only in Maritime Southeast Asia (Fig 4).

In mainland Southeast Asia, *Burkholderia pseudomallei*, *Orientia tsutsugamushi*, and *Rickettsia* antigens also featured in the rank lists for each scenario, as did *Leptospira* antigen in maritime Southeast Asia (Figs 1–3). While only *Rickettsia* was listed among the top five aetiologies of acute febrile illness in mainland Southeast Asia (Fig 4), the inclusion of these organisms is reflective of another major factor influencing analyte selection i.e., the prioritisation of pathogens which require specific directed antimicrobial therapy and have the potential to cause severe disease (Fig 5).

There was a clear preference for antigen-based pathogen biomarkers over serological analytes, including for pathogens like typhoidal *Salmonella* where such capillary blood-based

**Table 1. Regional panel participant demographic and professional background details.**

| Region | Characteristic | | | Round 1 (n, %) | Round 2 (n, %) |
|---|---|---|---|---|---|
| Mainland Southeast Asia [n = 19 (54%) and 23 (66%) in rounds 1 and 2, respectively] | Country | | Thailand | 2 (11) | 5 (22) |
| | | | Laos | 5 (26) | 5 (22) |
| | | | Cambodia | 3 (16) | 3 (13) |
| | | | Myanmar | 4 (21) | 4 (17) |
| | | | Peninsular Malaysia | 5 (26) | 6 (26) |
| | Gender | | Male | 15 (79) | 17 (74) |
| | | | Female | 4 (21) | 6 (26) |
| | Professional role | Clinical | Clinical infectious diseases | 10 (52) | 10 (43) |
| | | | Clinical infectious diseases and medical microbiology | 4 (21) | 5 (22) |
| | | | Other | 1 (5) | 2 (9) |
| | | Non-clinical | Medical microbiology | 3 (16) | 3 (13) |
| | | | Epidemiology | 2 (11) | 1 (4) |
| | | | Microbiology | 1 (5) | 2 (9) |
| | Years of experience | | 0 to 5 | 0 (0) | 1 (4) |
| | | | >5 to 10 | 1 (5) | 1 (4) |
| | | | >10 to 15 | 4 (21) | 5 (22) |
| | | | >15 to 20 | 3 (16) | 3 (13) |
| | | | >20 | 11 (58) | 13 (57) |
| Maritime Southeast Asia [n = 6 (29%) and 8 (38%) in rounds 1 and 2, respectively] | Country | | Indonesia | 2 (33) | 3 (38) |
| | | | Philippines | 1 (17) | 1 (13) |
| | | | Timor-Leste | 3 (50) | 4 (50) |
| | Gender | | Male | 1 (17) | 3 (38) |
| | | | Female | 5 (83) | 5 (62) |
| | Professional role | Clinical | Clinical infectious diseases | 5 (83) | 5 (63) |
| | | | Clinical infectious diseases and medical microbiology | 0 (0) | 1 (13) |
| | | | Other | 1 (17) | 2 (25) |
| | Years of experience | | >5 to 10 | 2 (33) | 3 (38) |
| | | | >10 to 15 | 3 (50) | 2 (25) |
| | | | >20 | 1 (17) | 3 (38) |
| South Asia [n = 11 (39%) and 13 (46%) in rounds 1 and 2, respectively] | Country | | India | 2 (18) | 2 (15) |
| | | | Nepal | 3 (27) | 4 (31) |
| | | | Pakistan | 4 (36) | 5 (38) |
| | | | Bangladesh | 2 (18) | 2 (15) |
| | Gender | | Male | 7 (64) | 7 (54) |
| | | | Female | 4 (36) | 6 (46) |
| | Professional role | Clinical | Clinical infectious diseases | 7 (64) | 7 (54) |
| | | | Clinical infectious diseases and medical microbiology | 1 (9) | 1 (8) |
| | | | Other | 1 (9) | 1 (8) |
| | | Non-clinical | Medical microbiology | 1 (9) | 2 (15) |
| | | | Public health | 0 (0) | 1 (8) |
| | | | Other | 1 (9) | 1 (8) |
| | Years of experience | | >10 to 15 | 0 (0) | 3 (23) |
| | | | >15 to 20 | 4 (36) | 3 (23) |
| | | | >20 | 7 (64) | 7 (54) |

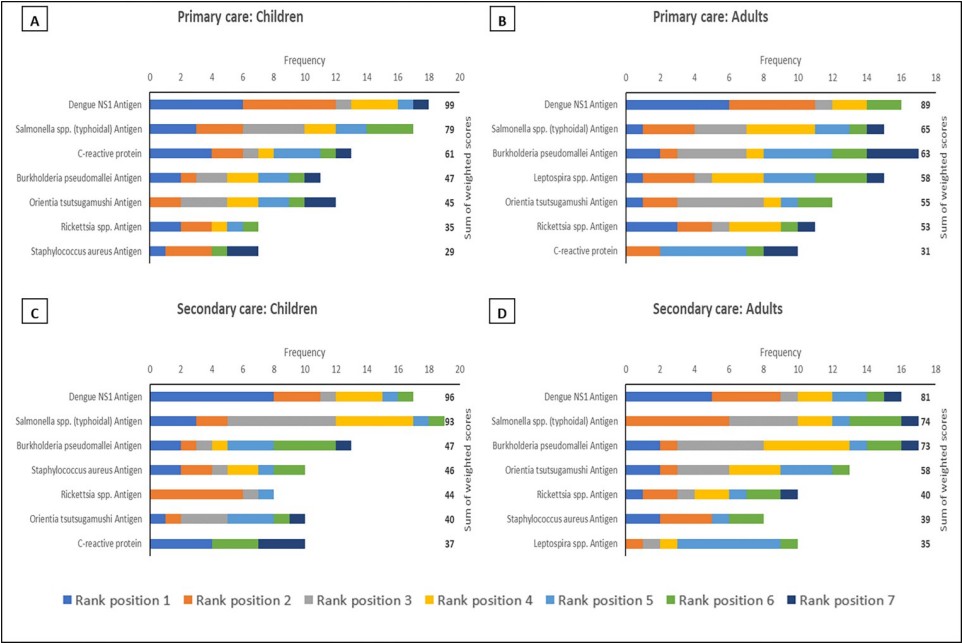

**Fig 1.** The top seven analytes, in descending order of priority based on sum of weighted scores, which should be considered for inclusion in multiplex lateral flow rapid diagnostic tests for acute non-malarial fever using fingerprick blood as the test substrate, in the following patient populations in Mainland Southeast Asia: (A) Children presenting to primary care settings (B) Adults presenting to primary care settings (C) Children being admitted to secondary care settings (D) Adults being admitted to secondary care settings. Children were defined as patients aged >28 days and <15 years. An analyte in rank position 1 was weighted seven times more than an analyte in rank position 7; the maximum sum of weighted scores per analyte was 161.

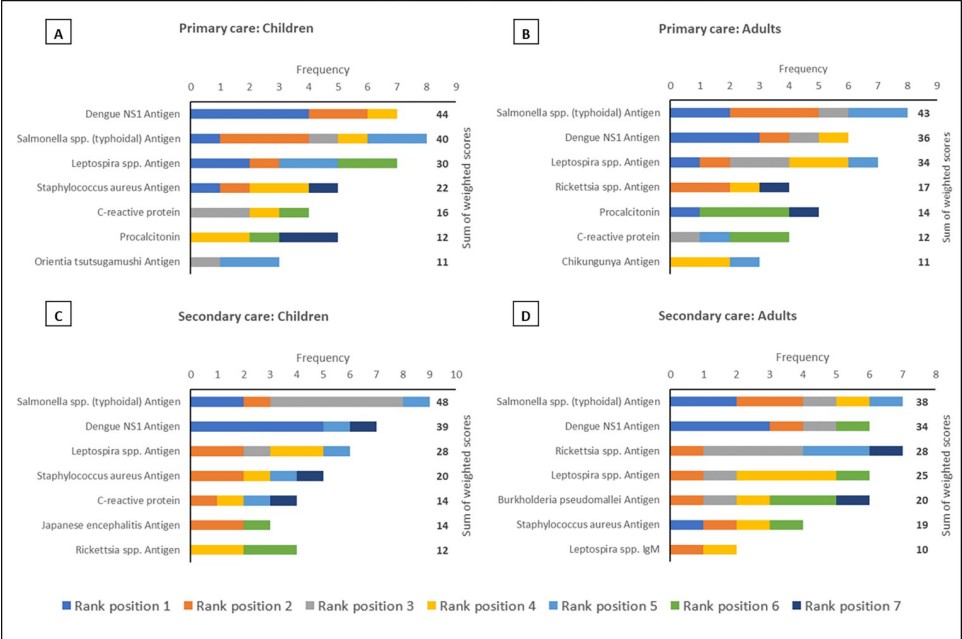

**Fig 2.** The top seven analytes, in descending order of priority based on sum of weighted scores, which should be considered for inclusion in multiplex lateral flow rapid diagnostic tests for acute non-malarial fever using fingerprick blood as the test substrate, in the following patient populations in Maritime Southeast Asia: (A) Children presenting to primary care settings (B) Adults presenting to primary care settings (C) Children being admitted to secondary care settings (D) Adults being admitted to secondary care settings. Children were defined as patients aged >28 days and <15 years. An analyte in rank position 1 was weighted seven times more than an analyte in rank position 7; the maximum sum of weighted scores per analyte was 56.

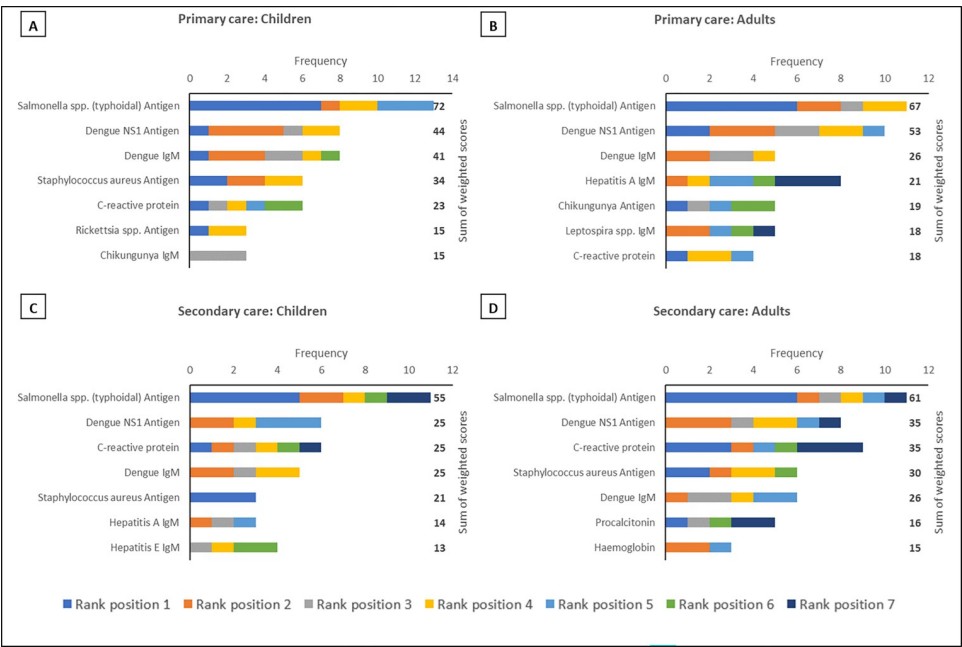

**Fig 3.** The top seven analytes, in descending order of priority based on sum of weighted scores, which should be considered for inclusion in multiplex lateral flow rapid diagnostic tests for acute non-malarial fever using fingerprick blood as the test substrate, in the following patient populations in South Asia: (A) Children presenting to primary care settings (B) Adults presenting to primary care settings (C) Children being admitted to secondary care settings (D) Adults being admitted to secondary care settings. Children were defined as patients aged >28 days and <15 years. An analyte in rank position 1 was weighted seven times more than an analyte in rank position 7; the maximum sum of weighted scores per analyte was 91.

rapid tests are not yet available (Figs 1–3). This finding also reflects the view of the majority of respondents that pathogens for which there are no readily available RDTs should be prioritised for inclusion when developing multiplex LF-RDTs (Fig 5).

The only host inflammatory biomarker which featured prominently in this survey was C-reactive protein (CRP). CRP was included in all rank lists except those for adult secondary care in Mainland and Maritime Southeast Asia (Figs 1–3). However, it was most frequently found in the bottom half of rank lists, indicative of the majority opinion that host biomarkers are less useful than microbiological diagnosis (Fig 5).

Furthermore, while twenty-eight (64%) respondents across all panels ranked CRP in one or more scenarios, there was little agreement on use case. Eight (29%) believed it was a good discriminator between viral and non-viral infections in all scenarios in which CRP featured in their rank lists, while two (7%) believed it to be a good marker of disease severity. A further eight (29%) thought CRP was able to fulfil both use cases, while 10 (36%) were of the opinion that use case varies depending on patient age and care setting.

For the scenario-based questions, the top-ranked three analytes in each rank list were unchanged between the first and second rounds although their individual positions varied slightly, except for the Maritime Southeast Asia and South Asia adult secondary care rank lists where the top-ranked two analytes were constant between rounds. The top-ranked analyte in all rank lists in the first round was either dengue NS1 antigen or typhoidal *Salmonella* antigen, demonstrating a degree of convergence at the end of the survey as both these analytes occupied the top two positions in all lists in the second round. The top three perceived aetiologies of acute febrile illness in all regions were also unchanged between rounds.

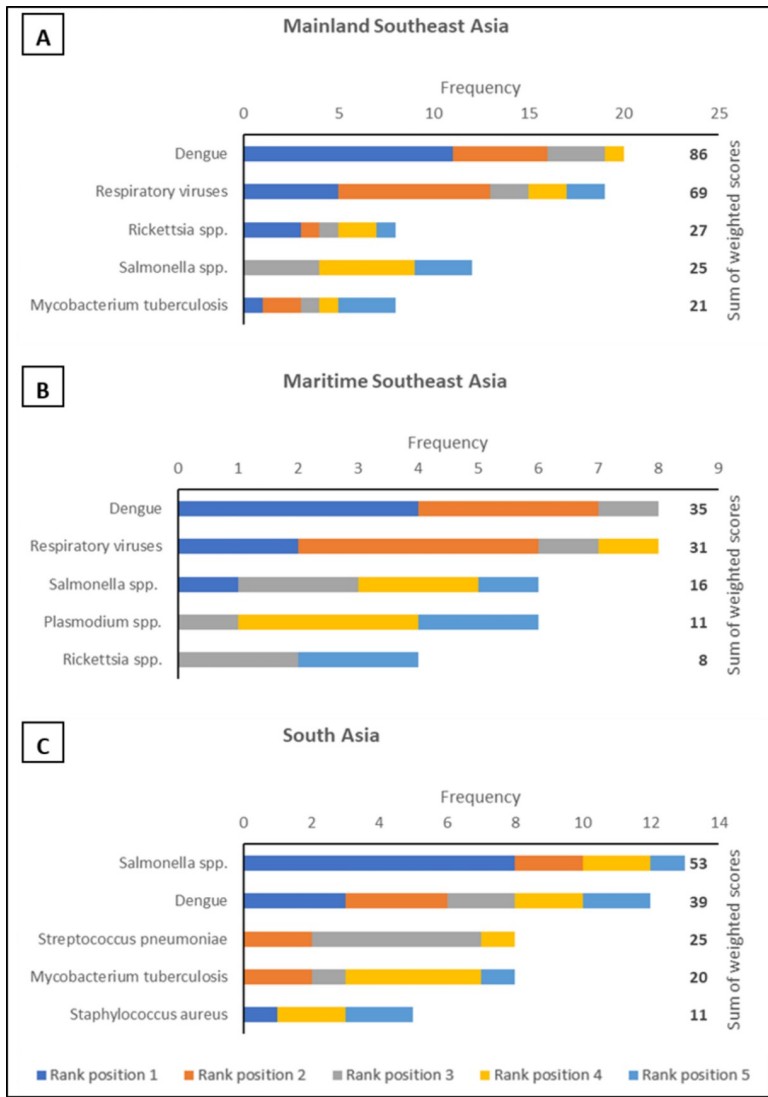

**Fig 4.** The top five commonest aetiologies of acute fever, in descending order of perceived annual incidence based on sum of weighted scores, in the following regions: (A) Mainland Southeast Asia (B) Maritime Southeast Asia (C) South Asia. An analyte in rank position 1 was weighted five times more than an analyte in rank position 5; the maximum sum of weighted scores per aetiology was 115 for Mainland Southeast Asia, 40 for Maritime Southeast Asia, and 65 for South Asia.

## Discussion

The results of this survey indicate that multiplex LF-RDTs for non-malarial acute febrile illness to be developed for use in South and Southeast Asia should be able to diagnose dengue and enteric fever in all age groups and care settings. There was also a clear desire for the development of LF-RDTs for pathogens not readily detected by existing technologies, and for these new tests to diagnose diseases directly through antigen detection. Pathogen biomarkers were prioritized over host biomarkers, for which the only contender for inclusion was CRP although there was little agreement on use case. The majority of the other pathogen biomarkers selected reflect the almost equal consideration given to clinical and epidemiological burdens of disease. Importantly, there was strong and early consensus that such LF-RDTs would aid clinical

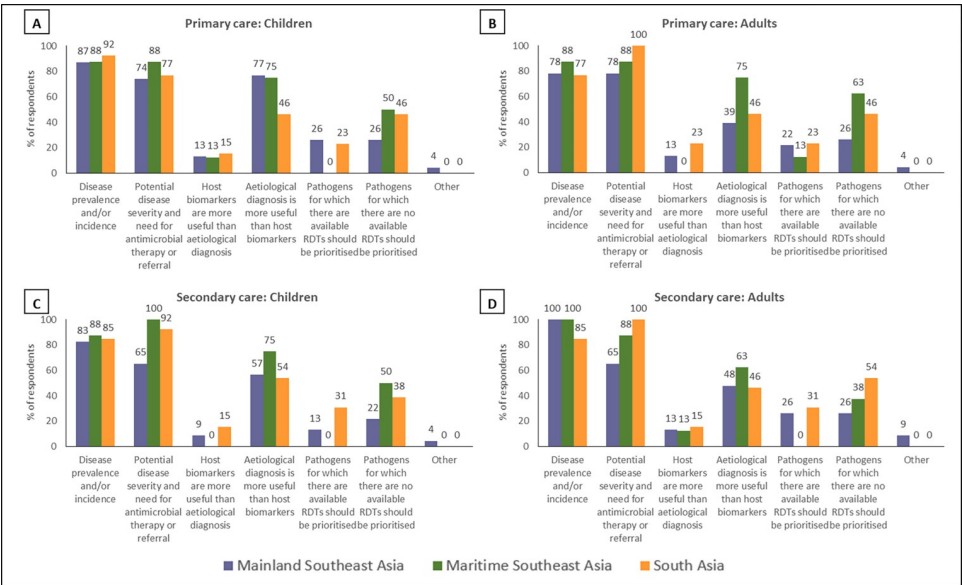

**Fig 5.** Reasons underlying the rankings made by individual expert participants of analytes which should be prioritized for inclusion in multiplex lateral flow rapid diagnostic tests for acute non-malarial fever using fingerprick blood as the test substrate, in the following patient populations: (A) Children presenting to primary care settings (B) Adults presenting to primary care settings (C) Children being admitted to secondary care settings (D) Adults being admitted to secondary care settings. Children were defined as patients aged >28 days and <15 years.

decision-making and that they should be incorporated into algorithm-based clinical decision support tools.

Our study has several strengths. Firstly, the modified Delphi method is well-suited to address the research question, given the challenges in collecting comprehensive epidemiological data before commencing work on LF-RDT development. The Delphi survey is a well-established technique facilitating consensus decision-making and avoiding domination by individual experts, as it is anonymous, systematic, iterative, and inclusive of a range of opinions [19,20]. Secondly, we adhered to best practice by pre-specifying the consensus definition, panellist selection criteria, and closing criterion, and by providing timely, well-structured controlled feedback [21,22]. In addition, we ensured generalizability through assembling appropriately-sized panels of suitably qualified members from diverse backgrounds and countries [21], and by taking steps to ensure that the response rate for each panel exceeded the approximately 30% response rate typical of online surveys [23]. Thirdly, the participant retention rate between the two rounds was high. However, by re-inviting panellists who did not participate in the first round, we avoided the possibility of false consensus and ensured optimal representation of perspectives [24]. Fourthly, although response stability was not a closing criterion, the consistency of the top-ranked two to three responses to each question between rounds is indicative of the reliability of our results [22,25].

The principal weakness of our study is that consensus, as per the prespecified definition, was not reached with regard to analyte rankings. However, we had anticipated this possibility and, thus, prespecified an alternative method of analyte prioritisation based on sums of weighted scores. Secondly, for pragmatic reasons we only surveyed experts from 12 out of the 15 LMICs in South and Southeast Asia. However, because panellist selection was weighted towards large countries both in terms of area and population, our results are still likely to be of considerable benefit in informing the development of LF-RDTs to be used in these regions.

Our results complement and extend previous work by Osborn et al. which, to our knowledge, is the only other study which has attempted to address this important question. They conducted a global prioritization exercise for a cartridge-based diagnostic test for pathogens causing severe febrile illness without a known source in patients presenting to secondary care [26]. Unlike their study, ours concentrated on acute non-malarial fever in both primary and secondary care settings with patients stratified by age, included host inflammatory biomarkers as options, and adopted a more granular regional focus. Both studies had typhoidal *Salmonella* as a top priority pathogen in addition to leptospirosis and rickettsioses, but we also recognized dengue and melioidosis as other major priorities given our emphasis on rural South and Southeast Asia. Additionally, for each pathogen we explored what participants deemed the ideal analyte, as well as the motivations and reasoning behind their choices. Our findings, therefore, add further layers of context which are essential to ensuring acceptability and applicability, two key challenges facing the development and implementation of any such LF-POCTs developed for use in these regions [27].

The obvious implication of this study is its value in informing analyte selection for multiplex LF-RDT development. In general, while current LF-RDT technologies do not yet permit the detection of more than 2–3 analytes from fingerprick blood, new diagnostic methods which allow detection of more analytes without an increase in sample volume are in development. Examples of these are CRISPR-, aptamer-, and SOMAmer-based multiplex assays which will also facilitate adaptation of the analyte panels, making them more context-specific [28,29]. The seven-item rank lists constructed will, thus, serve as a reference point as to which analytes to prioritise for inclusion in expanded panels in line with the target product profile for multiplex multi-analyte diagnostic platforms for acute febrile illness published by the World Health Organization [30].

Finally, we have also identified some key questions for future research. The first is the validation of setting-specific CRP use cases and the determination of cut-off levels to ensure sufficient sensitivity and specificity for each use case. The second is the development and validation of antigen-based LF-RDTs for typhoidal *Salmonella*, since no such tests have been developed despite the interest demonstrated by our study. The last concerns implementation research on contextual and other factors influencing the uptake of commercially available non-multiplex RDTs whose analytes were included in the rank lists produced through this study. It is essential to understand the reasons why CRP and dengue RDTs, for example, are not used as widely as would be expected and why variations in uptake between settings exist [31], because these may also be applicable to any multiplex LF-RDTs developed. It may also be the case that multiplexing may address some of the issues which have led to low uptake of single-plex assays, lending further support for multiplex LF-RDT development.

## Supporting information

**S1 Appendix. List of participating experts.**
(DOCX)

## Acknowledgments

We thank the experts who participated in this study, a list of whom can be found in S1 Appendix.

## Author Contributions

**Conceptualization:** Rusheng Chew, Yoel Lubell.

**Data curation:** Salisa Lohavittayavikant.

**Formal analysis:** Rusheng Chew.

**Funding acquisition:** Nicholas Philip John Day, Yoel Lubell.

**Investigation:** Rusheng Chew.

**Methodology:** Rusheng Chew, Yoel Lubell.

**Software:** Salisa Lohavittayavikant, Matthew Mayer.

**Supervision:** Nicholas Philip John Day, Yoel Lubell.

**Visualization:** Rusheng Chew.

**Writing – original draft:** Rusheng Chew.

**Writing – review & editing:** Salisa Lohavittayavikant, Matthew Mayer, Nicholas Philip John Day, Yoel Lubell.

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
