## [Decision Letter · Decision Letter 0]

4 Oct 2022

Dear Dr Chew,

Thank you very much for submitting your manuscript "Perceptions and priorities for the development of multiplex rapid diagnostic tests for acute non-malarial fever in rural South and Southeast Asia: an international modified e-Delphi survey" for consideration at PLOS Neglected Tropical Diseases. As with all papers reviewed by the journal, your manuscript was reviewed by members of the editorial board and by several independent reviewers. In light of the reviews (below this email), we would like to invite the resubmission of a significantly-revised version that takes into account the reviewers' comments. 

Please revise and provide a detailed response to each salient reviewer comment.

We cannot make any decision about publication until we have seen the revised manuscript and your response to the reviewers' comments. Your revised manuscript is also likely to be sent to reviewers for further evaluation.

Sincerely,

Georgios Pappas

Academic Editor

Justin Remais

Section Editor

Please revise accordingly to comments

Reviewer's Responses to Questions

**Key Review Criteria Required for Acceptance?**

**Methods**

-Are the objectives of the study clearly articulated with a clear testable hypothesis stated?

-Is the study design appropriate to address the stated objectives?

-Is the population clearly described and appropriate for the hypothesis being tested?

-Is the sample size sufficient to ensure adequate power to address the hypothesis being tested?

-Were correct statistical analysis used to support conclusions?

-Are there concerns about ethical or regulatory requirements being met?

Reviewer #1: The authors have attempted to develop a consensus through e-Delphi on prioritising analytes for inclusion in multiplex lateral flow rapid diagnostic tests (LF-RDTs) to diagnose non malarial fever in LMICs. The research question is relevant and important to address. Although the methodology is appropriate, the results lack any new insight or knowledge. The following are my specific comments: 

1. After two rounds, they failed to achieve the prespecified consensus of 80%. The alternative strategy to add the most common responses does not seem to have given any meaningful insight. The result does not add any new knowledge. 

2. The current common causes of acute febrile illness such as COVID-19 and influenza seem to have been totally overlooked. 

3. A lack of clear results would raise the question of soundness of the methodology. How were the participants selected? Were the four scenarios used appropriate and pretested to avoid ambiguity?

Reviewer #2: The objectives of the study are clearly articulated with a clear testable hypothesis stated.

The study design is appropriate to address the stated objectives.

The population is clearly described and appropriate for the hypothesis being tested.

The sample size is sufficient to ensure adequate power to address the hypothesis being tested.

Correct statistical analysis were used to support conclusions.

There are no concerns about ethical or regulatory requirements.

Reviewer #3: This manuscript describes a study to obtain expert consensus on priority analytes which should be included in multiplex lateral flow rapid diagnostic tests (LF-RDTs) for patients with acute non-malarial fever in South and Southeast Asian LMICs. The authors conducted two rounds of a modified e-Delphi survey with 84 panellists from 12 countries in three regions, representing Mainland Southeast Asia, Maritime Southeast Asia, and South Asia. There was consensus that LF-RDTs should be incorporated into clinical decision support tools. At a minimum, these LF-RDTs should be able to diagnose dengue and enteric fever in all patient categories. There was a clear preference to develop LF-RDTs for pathogens not readily detected by existing technologies, and for direct diagnosis through antigen detection versus serologic methods. Pathogen biomarkers were prioritized over host inflammatory biomarkers, with CRP being the only host biomarker ranked consistently highly. 

This is a well-designed and highly informative study. The objectives were clearly articulated and the study design is appropriate to address the study objectives. There are no ethical or regulatory concerns.

**Results**

-Does the analysis presented match the analysis plan?

-Are the results clearly and completely presented?

-Are the figures (Tables, Images) of sufficient quality for clarity?

Reviewer #1: 4. The findings could have been better presented. Figures 1, 2 and 3 could have been represented in a tabular format with percentages for a quick review as it’s difficult to read the graphs with too many colours. Figure 5 is well presented but it would help to add data labels (%) to each of the bars for better understanding and clarity.

Reviewer #2: The analysis presented match the analysis plan.

The results are clearly and completely presented.

The figures (Tables and Images) are of sufficient quality for clarity.

Reviewer #3: The analysis presented matched the analysis plan. The results are visually presented as a series of graphs that are easy to understand, despite their complexity. The figures are of sufficient quality for clarity.

**Conclusions**

-Are the conclusions supported by the data presented?

-Are the limitations of analysis clearly described?

-Do the authors discuss how these data can be helpful to advance our understanding of the topic under study?

-Is public health relevance addressed?

Reviewer #1: (No Response)

Reviewer #2: The conclusions are supported by the data presented.

The limitations of analysis are clearly described.

The authors discuss adequately, precisely and clearly how these data can be helpful to advance our understanding of the topic under study.

Public health relevance is addressed.

Reviewer #3: The conclusions are consistent with the findings of the study. The limitations of the study are clearly stated.

The authors discussed how the findings can be used to advance the diagnosis of febrile illness not caused by malaria.

**Editorial and Data Presentation Modifications?**

Reviewer #1: The introduction does not have a logical flow of thought. The opening statement of ‘Fever is a common but non-specific sign of sepsis’ itself is misleading. Sepsis commonly occurs secondary to a bacteraemic illness from a focus of infection such as pyelonephritis, pneumonia, intra-abdominal source, cellulitis, skin and soft tissue infection etc. The LF-RDTs are targeted towards tropical infections without a clear focus such as dengue, typhoid, scrub typhus / rickettsial infections, leptospirosis etc. The first paragraph confuses the goal with ‘sepsis’. Another example of lack of clarity is the sentence starting with Line101 which states ‘In addition, there is growing interest in augmenting pathogen-based diagnostics by assaying host biomarkers of inflammation in parallel, in particular those which can assist in the differentiation viral from non-viral infections, thus improving antimicrobial stewardship,[13] or predict or indicate severe disease, thus improving the identification of patients who require escalation of care.[14]’. Combining many ideas is confusing and made it very complex. The introduction requires substantial modification.

Reviewer #2: This paper represents an international modified e-Delphi survey that deals with the perceptions and priorities of selected infectious disease experts, who live and work in the area of South and Southeast Asia, for the development of acute non-malarial fever multiplex rapid diagnostic tests. The results of this study provide a valuable source of data on pathogen-specific biomarker analytes that should be incorporated in the test which would contribute significantly to rapid differential diagnosis of potential etiologic agents of acute non-malarial fever in resource-poor settings of above mentioned regions. In addition, the manuscript is clearly written, easy to read and understand. Given that there is very little data on this problem in the current literature, this study deserves to be published in “PLOS Neglected Tropical Diseases” journal. This reviewer has no additional suggestions.

Reviewer #3: (No Response)

**Summary and General Comments**

Reviewer #1: The authors have attempted to develop a consensus through e-Delphi on prioritising analytes for inclusion in multiplex lateral flow rapid diagnostic tests (LF-RDTs) to diagnose non malarial fever in LMICs. The research question is relevant and important to address. Although the methodology is appropriate, the results lack any new insight or knowledge.

Reviewer #2: (No Response)

Reviewer #3: Identification of regional priorities for clinical support of major syndromes is an important first step but studies have shown that availability of a diagnostic test is no guarantee that it will be used. This may be outside of the scope of this survey but it is important that the design of LF-RDTs for these priorities needs to take into account factors that will allow the tests to be widely deployed. If the authors have included in their survey questions related to the usability or challenges with current RDTs, it would be very useful to include them in the manuscript. Some examples for consideration by the authors are:

1. LF-RDTs for dengue IgM and NS1 are already commercially available. Since RDTs for dengue appear to be highly ranked in terms of priority, did the authors ask why they are not widely used? Do they need to have better performance or do they need to be multiplex to be useful (I.e. include other causes of febrile illness)?

2. Rapid tests for CRP have been available free of charge in some developed countries, such as UK, but uptake is less than 30%. Reasons for this have not been well studied but appear to be related largely to the difficulty of performing the test within the current patient pathway or requiring an additional follow-up visit. Did the authors include questions on how CRPs are currently used in either the public or private sector within the three regions?

3. Major causes of fever may vary by season and geographic location, even within the 3 regions of this study. This will affect not only the design of the multiplex LF-RDTs but also interpretation of the test results in terms of predictive values and likelihood ratios. It will be useful to include such data, if available.

PLOS authors have the option to publish the peer review history of their article (what does this mean?). If published, this will include your full peer review and any attached files.

Reviewer #1: No

Reviewer #2: No

Reviewer #3: Yes: Rosanna W Peeling
---

## [Editor Report · Decision Letter 1]

27 Oct 2022

Dear Dr Chew,

Thank you very much for submitting your manuscript "Perceptions and priorities for the development of multiplex rapid diagnostic tests for acute non-malarial fever in rural South and Southeast Asia: an international modified e-Delphi survey" for consideration at PLOS Neglected Tropical Diseases. As with all papers reviewed by the journal, your manuscript was reviewed by members of the editorial board. We are likely to accept this manuscript for publication, providing that you modify the manuscript according to the following recommendations. 

Only a minor change is needed, the rest of the responses are satisfying: Please further modify the statement on sepsis, even better omit it as a term. The manuscript is powerful enough to not need underlining of its significance with, strained I believe, correlations with sepsis.

Sincerely,

Georgios Pappas

Academic Editor

Justin Remais

Section Editor

Only a minor change is needed, the rest of the responses are satisfying: Please further modify the statement on sepsis, even better omit it as a term. The manuscript is powerful enough to not need underlining of its significance with, strained I believe, correlations with sepsis. Afterwards it will be immediately accepted

Figure Files:

Data Requirements:

Reproducibility:

References

---

## [Editor Report · Decision Letter 2]

28 Oct 2022

Dear Dr Chew,

We are pleased to inform you that your manuscript 'Perceptions and priorities for the development of multiplex rapid diagnostic tests for acute non-malarial fever in rural South and Southeast Asia: an international modified e-Delphi survey' has been provisionally accepted for publication in PLOS Neglected Tropical Diseases.

Best regards,

Georgios Pappas

Academic Editor

Justin Remais

Section Editor

Thank you for the revision

<style type="text/css">p.p1 {margin: 0.0px 0.0px 0.0px 0.0px; line-height: 16.0px; font: 14.0px Arial; color: #323333; -webkit-text-stroke: #323333}span.s1 {font-kerning: none

</style>

---

## [Editor Report · Acceptance letter]

7 Nov 2022

Dear Dr Chew,

We are delighted to inform you that your manuscript, "Perceptions and priorities for the development of multiplex rapid diagnostic tests for acute non-malarial fever in rural South and Southeast Asia: an international modified e-Delphi survey," has been formally accepted for publication in PLOS Neglected Tropical Diseases.

Best regards,

Shaden Kamhawi

co-Editor-in-Chief

Paul Brindley

co-Editor-in-Chief
